# Influence of Gender on Plasma Leptin Levels, Fat Oxidation, and Insulin Sensitivity in Young Adults: The Mediating Role of Fitness and Fatness

**DOI:** 10.3390/nu15112628

**Published:** 2023-06-04

**Authors:** Adrián Montes-de-Oca-García, Alejandro Perez-Bey, Juan Corral-Pérez, Alberto Marín-Galindo, Maria Calderon-Dominguez, Daniel Velázquez-Díaz, Cristina Casals, Jesus G. Ponce-Gonzalez

**Affiliations:** 1ExPhy Research Group, Department of Physical Education, Faculty of Education Sciences, University of Cadiz, 11519 Cadiz, Spain; adrian.montesdeoca@uca.es (A.M.-d.-O.-G.); juan.corral@uca.es (J.C.-P.); alberto.marin@uca.es (A.M.-G.); cristina.casals@uca.es (C.C.); jesusgustavo.ponce@uca.es (J.G.P.-G.); 2Unidad de Investigación, Instituto de Investigación e Innovación Biomédica de Cádiz (INiBICA), Hospital Universitario Puerta del Mar, 11009 Cadiz, Spain; alejandro.perezperez@uca.es (A.P.-B.); mariacalderond@gmail.com (M.C.-D.); 3GALENO Research Group, Department of Physical Education, Faculty of Education Sciences, University of Cadiz, 11519 Cadiz, Spain; 4Biomedicine, Biotechnology and Public Health Department, University of Cadiz, 11002 Cadiz, Spain; 5Advent Health Research Institute, Neuroscience Institute, Orlando, FL 32803, USA

**Keywords:** fatty acid oxidation, obesity, insulin resistance, cardiometabolic risk, physical fitness, dimorphism

## Abstract

It is unknown how plasma leptin affects fat oxidation depending on sex in young adults. Therefore, the present cross-sectional study aimed to examine the associations of plasma leptin with resting fat oxidation (RFO), maximal fat oxidation during exercise (MFO), and insulin sensitivity, considering the different responses in men and women, and the mediating role of fatness and cardiorespiratory fitness (CRF). Sixty-five young adults (22.5 ± 4.3 years; body mass index = 25.2 ± 4.7 kg·m^−2^, 23 females) participated in this study. Fasting plasma glucose, insulin, and leptin were analyzed. Variables related to insulin resistance (HOMA1-IR, HOMA2-IR), secretion (HOMA-%β), and sensitivity (HOMA-%S, QUICKI) were computed. RFO and MFO were determined through indirect calorimetry. A peak oxygen uptake (VO_2_peak) test was performed until exhaustion after the MFO test. The MFO was relativized to body mass (MFO-BM) and the legs’ lean mass divided by the height squared (MFO-LI). In men, leptin was negatively associated with MFO-BM and positively with HOMA-%β (*p* ≤ 0.02 in both). In women, leptin was positively associated with RFO and QUICKI, and negatively with MFO-BM (*p* < 0.05 in all). The association between leptin and MFO was mediated by CRF (*p* < 0.05), but not by fat mass (*p* > 0.05). Plasma leptin is associated with fat oxidation and insulin secretion/sensitivity, with different responses within each sex. The association between leptin and fat oxidation is mediated by cardiorespiratory fitness.

## 1. Introduction

Cardiometabolic diseases such as obesity or type 2 diabetes mellitus are currently one of the main health problems that humans face [1,2]. In fact, more and more people are suffering from metabolic inflexibility, which is partly characterized by reduced fat oxidation by the skeletal muscles under fasting conditions and in the postprandial state [3]. Moreover, healthy young adults with an impaired ability to oxidize fat during exercise (i.e., a decreased maximal fat oxidation capacity), which appears to be related, among others, to genetic factors [4], are at increased cardiometabolic risk [1].

Another reason for metabolic inflexibility is the breakdown of homeostasis at the cellular level due to the resistance of peripheral tissues to fundamental hormones such as insulin [5], produced and secreted by the beta cells of the islets of Langerhans of the pancreas, and leptin [6], an adipocytokine secreted by the adipose tissue which has a regulatory role in the central nervous system [7]. Likewise, it has been shown that one of the main causes of hyperleptinemia is obesity, which in turn causes leptin resistance due to lipotoxicity [6]. Moreover, leptin resistance affects, among others, the feeling of appetite, which can aggravate the obese phenotype by worsening body composition, leading to increased hyperphagia, hyperglycemia, hyperinsulinemia, and insulin resistance [8]. Furthermore, it is known that leptin promotes satiety and reduces caloric intake by either stimulating pro-opiomelanocortin (POMC) neurons in the arcuate nucleus (ARC) of the hypothalamus or inhibiting their inhibitors (NPY/AgRP neurons), leading to appetite suppression [9].

On the other hand, it has been recently observed that high cardiorespiratory fitness (CRF) is related to lowering circulating leptin levels [10], which generally translates into greater leptin sensitivity and higher fat oxidation. In this sense, Antunes et al. [11] reported that individuals with low CRF exhibit some metabolic–endocrine disruptions, such as hyperleptinemia, impaired lipid profile, and low levels of adiponectin, which translates into a worse health status. In fact, CRF plays a protective role as a moderator in the relationship between obesity and adipocytokines such as leptin and adiponectin, suggesting that people should be engaged in physical activity to improve CRF levels and consequently improve cardiometabolic health [12]. Equally, there is a strong association between improved CRF and reduced insulin resistance, which results in a reduced risk for prediabetes or type 2 diabetes mellitus [13]. Thus, CRF could play a fundamental role in leptin and insulin signaling and, consequently, in fat oxidation and insulin sensitivity.

Nevertheless, it is not clear which of these two factors (fatness or fitness) has greater importance in leptin and insulin signaling and fat oxidation at rest and during exercise in healthy young people. Moreover, it is also unknown how leptin affects fat oxidation depending on sex in this population, although it is known that there is gender dimorphism in skeletal muscle leptin receptors expression, which can be partly explained by the influence of testosterone, with higher leptin levels in women [14]. In fact, higher rates of fat oxidation have been observed in women, regardless of CRF [15], although it is unknown whether these differences could be due to higher leptin levels in women. Hence, this study aimed to investigate the associations of plasma leptin with resting fat oxidation (RFO), maximal fat oxidation (MFO) during exercise, and insulin sensitivity in young adults, considering the different responses in men and women, and the mediating role of fatness and fitness. The hypothesis was that plasma leptin levels are associated with fat oxidation and insulin sensitivity, with different responses within each sex, and that both body fat mass and physical fitness have a mediating effect on this association.

## 2. Materials and Methods

### 2.1. Design

This cross-sectional study is part of the NutAF research project [1,4,16,17]. The study was approved by the Ethical Committee of the Hospital Puerta del Mar (Cadiz, Spain), according to the Declaration of Helsinki. All tests were performed in the Laboratory of Physical Activity and Exercise of the University of Cadiz (Puerto Real, Cádiz, Spain). Written informed consent was obtained by all the participants after being informed about the nature of the study, the protocols, and the possible risks arising from the measurements.

### 2.2. Subjects

From the total sample of the NutAF study, a cohort of 65 participants (22.55 ± 4.30 years; body mass index (BMI) = 25.26 ± 4.74 kg·m^−2^, *n* = 23 females) with complete data of leptin concentration, fat mass, CRF, RFO, MFO, and glycemic parameters were enrolled in the study. All participants were young adults without known diseases but with different profiles of body composition. The inclusion criteria were being between 18 and 45 years old, having a stable body mass (±2 kg) during the last six months, and BMI between 18.5 and 40 kg·m^−2^. The exclusion criteria were having made a weight loss diet or a specific diet different from the regular one during the last six months or suffering any illness or injury that prevented physical exercise. These inclusion and exclusion criteria were selected because it has been observed in the scientific literature that changes in body mass induced by exercise and diet can modify the fat oxidation capacity and insulin sensitivity in young adults [18].

### 2.3. Procedure

On the day of measurements, blood samples and laboratory tests were performed in the morning in a fasting situation. Participants were instructed to maintain their usual diet and hydration and to avoid alcohol and caffeine intake and intense physical activity the day before. All the measurements were performed in a conditioned room (21 ± 1 °C, 50 ± 2% relative humidity). Firstly, blood samples were obtained to determine blood glucose, insulin, and leptin. After that, anthropometric and body composition measurements were performed. Finally, indirect calorimetry was performed for the analysis of resting metabolism (RFO) and the exercise protocol to determine MFO and CRF, expressed as peak oxygen uptake (VO_2_peak).

#### 2.3.1. Blood Extraction and Biochemical Parameters

Fasting blood samples were taken from the antecubital vein and collected in EDTA tubes which were centrifuged (2500 rpm, 15 min, 4 °C) to obtain plasma, which was stored at −80 °C until analyses. Blood glucose was measured using a commercial kit from Spinreact (glucose-HK, ref. 1001200) and following the manufacturer’s instructions. The intra-assay coefficient of variation was <1% and the inter-assay coefficient of variation was <1.5%. Absorbances were obtained using a BIO-TEK PowerWaveTM 340 microplate reader and the BIO-TEK KC JuniorTM program (Bio-Tek Instruments Inc., Winooski, VT, USA). Additionally, plasma insulin and leptin levels were measured using MILLIPLEX^®^ MAP Human Metabolic Hormone Magnetic Bead Panel (HMHEMAG-34K, Millipore Sigma, Burlington, MA, USA) and Luminex^®^ 200TM System (Luminex Corp., Austin, TX, USA) according to the manufacturer’s instructions. The intra-assay coefficients of variation were <15% for both insulin and leptin.

From the insulin data in pg·mL^−1^, a conversion to mUI·L^−1^ was made [19] and the HOMA-1IR (homeostasis model assessment of insulin resistance) and the QUICKI (quantitative insulin sensitivity check index) were calculated as a previous study of the NutAF project described [20]. HOMA-IR was calculated according to the following formula: HOMA-IR = fasting blood glucose (mg·dL−^1^) × fasting insulin (mUI·L^−1^)/405. QUICKI was calculated according to the following formula: QUICKI = 1/[log (fasting insulin mUI·L^−1^) + log (fasting blood glucose mg·dL^−1^)]. Moreover, the HOMA2-IR (including data from the percentage of steady state beta cell function (insulin secretion), HOMA2-%β, and percentage of insulin sensitivity, HOMA2-%S) was obtained by the program HOMA Calculator v2.2.3 (https://www.dtu.ox.ac.uk/homacalculator/ (accessed on 15 October 2022)).

#### 2.3.2. Anthropometry and Body Composition

Height was measured in a standing position, after normal expiration, using a height rod (SECA 225, range from 60 to 200 cm; the precision of 1 mm). Body mass, body fat, and lean body mass were evaluated using a multi-frequency bioimpedance of 8 electrodes (TANITA-MC780MA, Barcelona, Spain). BMI was calculated as the body mass divided by the height squared (kg·m^−2^). The subjects wore light clothing and adopted a specific posture according to the manufacturer’s instructions. Additionally, the participants were required to urinate before measurement to ensure the elimination of body fluids. Moreover, the measurement was carried out without metallic objects in the body that could alter the results.

#### 2.3.3. Resting Metabolism

Oxygen uptake (VO_2_) and carbon dioxide production (VCO_2_) were registered in resting conditions lying on a bed in a supine position for 30 min for calculating the respiratory exchange ratio (RER) and RFO in a conditioned room (21 ± 1 °C, 50 ± 2% relative humidity). A mask was placed on the subject’s face to collect gas samples. An open-circuit gas analyzer (Jaeger MasterScreen CPX^®^ CareFusion, San Diego, CA, USA) was used to register indirect calorimetry data. Calibrations were performed daily before each measurement. During the test, the gas analyzer values were captured breath-by-breath and averaged every 20 s. For the analysis of these variables, the first 5 min of the evaluation were eliminated and a stable period of 5 min was selected with a coefficient of variation for VO_2_ and VCO_2_ lower than 15%. The average values of VO_2_ and VCO_2_, in the selected time interval, were used to calculate the resting metabolism (kcal) by an indirect equation proposed by Frayn [21].

#### 2.3.4. Maximal Fat Oxidation (MFO) and Cardiorespiratory Fitness (VO_2_peak)

An incremental protocol on a cycle ergometer (Lode Excalibur, Groningen, The Netherlands) was designed from the standardized protocol [22], with two consecutive phases to determine MFO and VO_2_peak. For the determination of MFO, the first phase consisted of 3 min steps with 15 W increments in overweight/obese subjects and 30 W in normal weight subjects, with a maintained pedaling rate between 60–80 rpm. This phase was interrupted when RER ≥ 1.0. After a brief pause (between 3 and 5 min), the second phase to detect VO_2_peak was initiated. This phase began at the load at which phase 1 ended and continued with 1 min steps increasing at the same load rate as in phase 1, with an equal cadence. This phase ended when the participant reached exhaustion. The protocol was considered maximum when the VO_2_ reached a plateau, the theoretical maximal heart rate was reached, and when RER ≥ 1.10. When the maximality criteria were not met, CRF was defined as the value of VO_2_peak. RER, VO_2_, and VCO_2_ were measured by indirect calorimetry (Jaeger MasterScreen CPX^®^). To calculate the fatty acid oxidation in the different steps of the protocol, the average values of VO_2_ and VCO_2_ were used in the last 60 s of each step of the test, applying the Frayn equation [21]. Similarly, the average value of VO_2_ was used to determine the % VO_2_peak reached in each step. With the values obtained from fat oxidation and % VO_2_peak in each step, a polynomial curve that best fits the results of the present analysis was drawn for each participant [23]. Finally, the absolute value of MFO was relativized to different values of body composition to create normalized variables: MFO relativized to the total body mass (MFO-BM) and MFO legs index, which was relativized to the legs’ lean mass divided by the height squared (MFO-LI) [1].

### 2.4. Statistical Analyses

According to one of our main variables, MFO, the sample size was calculated using the G*Power software (v. 3.1.9.7, University of Kiel, Kiel, Germany), with a statistical power of 0.96, an effect size of 0.82, and a significance level of 0.05. For that, the required sample size was at least 23 subjects. Parametric tests were performed since the normal distribution of the variables was previously verified by the Kolmogorov–Smirnov test. Descriptive statistics were calculated and expressed as mean ± standard deviation (SD), and the comparisons between sexes were determined through a Student *t*-test. To verify the extent to which leptin is related to fat oxidation (RFO and MFO), insulin resistance (HOMA1-IR and HOMA2-IR), insulin secretion (HOMA-%β), and insulin sensitivity (HOMA-%S and QUICKI), linear regression analyses were performed by including each of these variables as outcomes. Four different linear regression models were performed: unadjusted (model 1) and adjusted by age (model 2), fat mass (model 3), and CRF (model 4). Models 3 and 4 were relativized in the same way as the dependent variable (for example, if the dependent variable was relativized to body mass, so was the adjusting variable). Moreover, since it has been shown that there is sexual dimorphism in the expression of leptin, with higher levels of leptin in women [14], all linear regression analyses were separated by sex. Further, in the present study, this statement was confirmed with a previous sex interaction analysis of leptin concentration with body fat percentage using linear regression analysis (*p* < 0.001). Further, to verify the extent to which fat mass and CRF could be influencing these associations, mediation analyses were carried out including fat mass and CRF as mediating variables. All mediating variables were also relativized in the same way as the dependent variable. Linear regression models were fitted using the bootstrapping analysis by Hayes’ PROCESS macro for SPSS (Armonk, NY, USA), using a resampling procedure of 10,000 bootstrap samples [24,25]. This method aims to assess the total effects (Equation (2)) and direct effects (Equations (1), (3) and (3′)) depicted by the unstandardized regression coefficient and significance among the independent and dependent outcomes. Equation (1) regressed the independent variable (leptin) on the mediators (CRF and fat mass, separately). Equation (2) regressed the independent variable on the dependent variables (RFO, MFO, HOMA, and QUICKI). Equation (3) regressed both mediators (Equation (3)) and the independent variables (Equation (3′)) on the dependent variables. The indirect effects and confidence intervals (CI) were also included. Mediation was accepted when the indirect effect was significantly different from zero. Moreover, to establish mediation, Equations (1) and (3) should be significant as well as the indirect effect. As in the linear regression analyses, gender was a covariate in all mediation analyses. Effect size statistics were calculated, with Cohen’s d for the Student *t*-test and R^2^ (coefficient of determination) for linear regression analysis. The significance was set at *p* < 0.05.

## 3. Results

### 3.1. Descriptive Statistics and Differences by Sex

The general characteristics of the participants and differences between men and women are shown in Table 1. Differences were found in height (higher in men; *p* < 0.001), body mass (higher in men; *p* = 0.039), total lean mass (higher in men; *p* < 0.001), body fat percentage (higher in women; *p* < 0.001), VO_2_peak (higher in men; *p* = 0.001), plasma leptin (higher in women; *p* < 0.001), blood glucose (higher in men; *p* = 0.036), and MFO-LI (higher in women; *p* = 0.023).

### 3.2. Associations of Leptin Concentration with Fat Oxidation and Insulin Sensitivity in Men and Women

In men (Table 2), leptin was associated with MFO-BM and HOMA-%β. In model 1 (unadjusted), leptin was negatively associated with MFO-BM (β = −0.359; R^2^ = 0.129; *p* = 0.020) and positively associated with HOMA-%β (β = 0.398; R^2^ = 0.158; *p* = 0.009). In model 2 (adjusted for age), leptin was positively associated with HOMA-%β (β = 0.430; R^2^ = 0.175; *p* = 0.007). No significant associations were found in model 3 (adjusted for fat mass). Finally, in model 4 (adjusted for CRF) leptin was positively associated with HOMA-%β (β = 0.469; R^2^ = 0.167; *p* = 0.015).

In women (Table 2), leptin was associated with RFO, MFO-BM, and QUICKI. In model 1 (unadjusted), leptin was positively associated with RFO (β = 0.501; R^2^ = 0.251; *p* = 0.015) and negatively associated with MFO-BM (β = −0.436; R^2^ = 0.190; *p* = 0.037). In model 2 (adjusted for age), leptin was positively associated with RFO (β = 0.584; R^2^ = 0.311; *p* = 0.007). In model 3 (adjusted for fat mass), leptin was positively associated with RFO (β = 0.807; R^2^ = 0.296; *p* = 0.023). Finally, in model 4 (adjusted for CRF) leptin was positively associated with RFO (β = 0.538; R^2^ = 0.271; *p* = 0.013) and QUICKI (β = 0.597; R^2^ = 0.181; *p* = 0.049).

### 3.3. Mediation Analyses: Adiposity and CRF

Figure 1 shows the mediation models of fat mass in the associations between circulating leptin and RFO, absolute MFO, MFO-BM, and MFO-LI, whereas Figure 2 shows the mediation models of CRF. Likewise, Figure 3 shows the mediation models of fat mass in the associations between circulating leptin and HOMA1-IR, HOMA2-IR, HOMA-%β, HOMA-%S, and QUICKI, whereas Figure 4 shows the mediation models of CRF.

Statistically significant mediating effects were found in MFO-BM and MFO-LI. Concerning MFO-BM, the relationship with leptin was mediated by CRF, but not by fat mass. In Equation (1), leptin was negatively associated with CRF (B = −0.001; β = −0.680; *p* < 0.001). In Equation (2), leptin was negatively associated with MFO-BM (B = −0.001; β = −0.438; *p* = 0.003). In Equation (3), CRF was positively associated with MFO-BM (B = 0.210; β = 1.020; *p* < 0.001). Finally, in Equation (3′), leptin was positively associated with MFO-BM (B = 0.001; β = 0.256; *p* = 0.027). Moreover, the indirect effect of CRF was statistically significant (*p* < 0.05).

Likewise, the relationship between leptin and MFO-LI was not mediated by fat mass, but it was partially mediated by CRF. In Equation (1), leptin was negatively associated with CRF (B = −0.009; β = −0.372; *p* = 0.005). In Equation (2), leptin was negatively associated with MFO-LI (B = −0.001; β = −0.312; *p* = 0.031). In Equation (3), CRF was positively associated with MFO-LI (B = 0.015; β = 0.612; *p* < 0.001). Finally, in Equation (3′), the association between leptin and MFO-LI was not statistically significant (B = −0.001; β = −0.084; *p* = 0.505). Therefore, although the indirect effect of CRF was statistically significant (*p* < 0.05), it cannot be considered a pure mediating effect of CRF, since Equation (3′) was not statistically significant (inconsistent mediation).

The mediation models of fat mass in the associations between circulating leptin levels and RFO, absolute MFO, MFO-BM, and MFO-LI are shown. The statistically significant indirect effect appears in bold. All mediating variables were relativized in the same way as the dependent variable. Abbreviations: *p*, *p*-value; B, non-standardized coefficient; β, standardized coefficient; CI, confidence interval; RFO, resting fat oxidation; MFO, maximal fat oxidation during exercise; MFO-BM, MFO relativized to body mass; MFO-LI, MFO relativized to the legs’ lean mass divided by the height squared; Fat mass-BM, fat mass relativized to body mass; Fat mass-LI, fat mass relativized to the legs’ lean mass divided by the height squared.

The mediation models of fat mass in the associations between circulating leptin levels and RFO, absolute MFO, MFO-BM, and MFO-LI are shown. The statistically significant indirect effect appears in bold. All mediating variables were relativized in the same way as the dependent variable. Abbreviations: *p*, *p*-value; B, non-standardized coefficient; β, standardized coefficient; CI, confidence interval; RFO, resting fat oxidation; MFO, maximal fat oxidation during exercise; MFO-BM, MFO relativized to body mass; MFO-LI, MFO relativized to the legs’ lean mass divided by the height squared; CRF-BM, CRF relativized to body mass; CRF-LI, CRF relativized to the legs’ lean mass divided by the height squared.

Regarding the rest of the variables included in the analyses (HOMA1-IR, HOMA2-IR, HOMA-%β, HOMA-%S, and QUICKI), no statistically significant mediating effects were found (*p* > 0.05).

The mediation models of fat mass in the associations between circulating leptin and HOMA1-IR, HOMA2-IR, HOMA-%β, HOMA-%S, and QUICKI are shown. The statistically significant indirect effect appears in bold. All mediating variables were relativized in the same way as the dependent variable. Abbreviations: *p*, *p*-value; B, non-standardized coefficient; β, standardized coefficient; CI, confidence interval; HOMA-IR, homeostasis model assessment of insulin resistance; HOMA-%β, percentage of steady state beta cell function (insulin secretion); HOMA-%S, percentage of insulin sensitivity; QUICKI, quantitative insulin sensitivity check index.

The mediation models of CRF in the associations between circulating leptin and HOMA1-IR, HOMA2-IR, HOMA-%β, HOMA-%S, and QUICKI are shown. The statistically significant indirect effect appears in bold. All mediating variables were relativized in the same way as the dependent variable. Abbreviations: *p*, *p*-value; B, non-standardized coefficient; β, standardized coefficient; CI, confidence interval; HOMA-IR, homeostasis model assessment of insulin resistance; HOMA-%β, percentage of steady state beta cell function (insulin secretion); HOMA-%S, percentage of insulin sensitivity; QUICKI, quantitative insulin sensitivity check index; CRF-BM, CRF relativized to body mass.

## 4. Discussion

### 4.1. Main Research Findings

The results of the present study suggest that leptin correlates negatively with fat oxidation during exercise and positively with insulin secretion in men, while leptin negatively relates to fat oxidation during exercise and positively with resting fat oxidation and insulin sensitivity in women. Fitness appeared to mediate the exercise fat oxidation results. Together, although leptin is linked to fat mass, these results suggest physical fitness contributes independently of fat mass on influencing leptin’s relation to fuel use. Therefore, the present study not only confirms that there is sexual dimorphism in leptin expression, with higher levels of plasma leptin in women [14], but also that leptin affects fat oxidation capacity and insulin sensitivity differently in men and women. In fact, the findings of the present study suggest that both gender and CRF play a key role in the relationship between the concentration of leptin, fat oxidation, and insulin sensitivity in young adults.

### 4.2. Association between Plasma Leptin and RFO

In our study, leptin was positively associated with RFO only in women, regardless of age, fatness, and CRF, and it could be explained by the higher levels of estrogens in women, which implies an increase in the oxidative capacity of the skeletal muscle both in a resting situation and during exercise [26]. However, this different response was not found in MFO during exercise, since it depends mainly on CRF [27], among other factors. In addition, the association between circulating leptin and RFO was maintained, and its strength even increased, when CRF was considered. This fact suggests that elevated leptin levels do not result in a reduced RFO, but, on the contrary, it increases this capacity. In fact, in a previous study [28] it was observed that RFO was positively associated with resting metabolism, and these results coincide with the results of another previous study in which it was shown that resting metabolism is not altered in people with obesity since leptin concentration was positively associated with resting energy expenditure [29]. Thus, it seems that in a resting situation, when a low rate of energy production is needed, the peripheral tissues are sensitive enough to leptin to promote normal functioning of oxidative metabolism, especially if the subjects have a high CRF level. Nonetheless, these findings contradict the results of a previous study in which it was shown that the hyperleptinemia state can downregulate energy expenditure in obese women [30]. However, the sample characteristics, in terms of the CRF level or other factors that affect the fat oxidation capacity, such as an impaired mitochondrial function, could justify these differences [31].

### 4.3. Association between Plasma Leptin and MFO: The Mediating Role of CRF and Adiposity

Both in men and women, our data showed an inverse and independent relationship between leptin and MFO relativized to body mass, although the association was lost when age, fat mass, and CRF were considered. Interestingly, when CRF was included in the mediation analysis, leptin was positively associated with MFO values, suggesting that at equal CRF, higher leptin levels are associated with higher MFO during exercise. This seems logical knowing that CRF affects the leptin concentration by decreasing the levels of this hormone and improving the leptin sensitivity of peripheral tissues [32]. Thus, for the same leptin sensitivity and CRF, higher concentrations of this hormone seem to be associated with higher fatty acid oxidation capacity. Consistent with our results, a study by Ara et al. [33] demonstrated that obese subjects, who have higher levels of leptin, have a normal mitochondrial function and higher MFO compared to normal weight subjects, both with similar CRF levels, suggesting that CRF plays a key role in the relationship between leptin and fat oxidation. In fact, our study showed that CRF has a significant indirect effect on the relationship between leptin and MFO, demonstrating that when CRF is considered, leptin has a positive effect on MFO capacity. Therefore, our study supports the importance of physical fitness in metabolic health, since physical fitness can modulate the leptin signaling pathway in the skeletal muscle [34]. Nevertheless, it should be noted that in our study the inclusion of fat mass did not imply a significant change in the direction of the associations. Therefore, according to our findings, it could be stated that CRF has more impact than fatness in the relationship between leptin concentration and fat oxidation. However, a recent study by Frandsen et al. [35] showed that unfit women with obesity, regardless of age, have a higher MFO compared to unfit normal weight women, suggesting that adiposity also plays a key role in fat oxidation. Thus, it appears that both fatness and CRF affect the fat oxidation capacity, which in turn is influenced by leptin. Nonetheless, it must be considered that fat oxidation in young adults is not only influenced by age, gender, adiposity, or physical fitness but is also influenced by the levels of physical activity and sedentary behavior [17], although this was not taken into account in our design.

### 4.4. Association between Plasma Leptin and Insulin

Additionally, in our study leptin concentration was positively associated with insulin secretion (HOMA-%β) only in men, although the association was lost when fat mass was considered, suggesting that those subjects with higher levels of leptin have greater functionality of the insulin-producing pancreatic β-cells, which can lead to resistance to this hormone by peripheral tissues. This association was also significant when age and CRF were considered, but not when body fat mass was considered. Therefore, the results of previous studies in which higher levels of leptin are related to higher levels of insulin [36] and, consequently, insulin resistance [8,37] due to a break in the insulin signaling pathway caused by hyperinsulinemia are supported. In fact, it is known that insulin resistance can be exacerbated by the increase in free fatty acids flux, hyperleptinemia, and hyperglycemia, causing lipotoxicity and glucotoxicity, leading to β-cells failure and eventually type 2 diabetes mellitus [7]. Moreover, in our study leptin concentration was positively associated with QUICKI only in women when CRF was considered in the analysis, which suggests that at the same CRF level, higher levels of leptin in women lead to greater sensitivity to insulin. In fact, these differences could be justified due to the higher levels of estrogens in premenopausal women compared to men, which confer a cardioprotective effect [26]. Thus, the results of the present study suggest that women have a lower cardiovascular risk and a lower risk of developing metabolic diseases such as type 2 diabetes mellitus, and they also show that high leptin levels appear to be positive in reducing this risk if adequate levels of physical fitness are maintained. In fact, despite adiposity, obese people can be metabolically healthy if they maintain high levels of physical activity and fitness that help maintain homeostasis at the cellular level [38].

### 4.5. Limitations, Strengths, and Future Lines of Research

Finally, some limitations should be taken into consideration. Firstly, the sample was small, composed of young adults, and with an unbalanced male-to-female ratio. Therefore, the extrapolation of results should take this fact into account. The measurement of body composition by bioimpedance instead of using more precise methods such as dual-energy X-ray absorptiometry (DXA) is a limitation too. Additionally, the phases of the menstrual cycle and the estrogen levels in women were not considered, despite their great influence on fat oxidation and insulin sensitivity [26]. Moreover, we used indirect equations to estimate insulin resistance as opposed to using the euglycemic hyperinsulinemic clamp with stable isotopes, and indirect calorimetry to determine fat oxidation instead of using more objective and direct methods. In addition, the diet of the day before was not controlled, and this could affect the oxidation of energy substrates during the day of the measurements. Notwithstanding, the determination of fat oxidation and cardiorespiratory fitness in a controlled situation (with a laboratory test, gas analyzer, etc.), and the inclusion of biochemical parameters bring strengths to the design. We also consider a strength of our study to have a sample of men and women with similar fat oxidation levels, despite the differences in physical fitness and adiposity. Nonetheless, experimental designs in which it is investigated whether exercise-induced changes in the leptin signaling pathway at the biomolecular level are correlated to changes in fat oxidation or insulin sensitivity, considering gender differences, are warranted.

## 5. Conclusions

Circulating leptin levels are associated with fat oxidation and insulin secretion/sensitivity, with different responses in men and women. Further, the association between plasma leptin and MFO appears to be mediated by CRF, but not by body fat mass, highlighting the indirect effect of physical fitness in the function exerted by leptin on fat oxidation. These results confirm that physical fitness plays a key role in the metabolic health of young adults.

## Figures and Tables

**Figure 1 nutrients-15-02628-f001:**
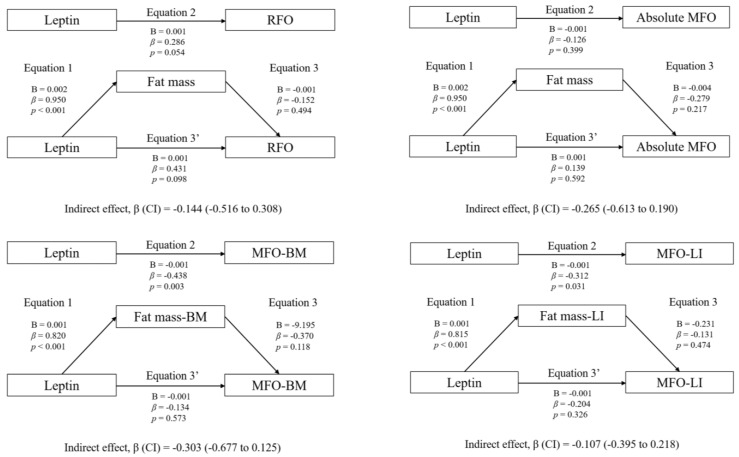
Mediation analysis of fat mass in the association between leptin concentration and fat oxidation.

**Figure 2 nutrients-15-02628-f002:**
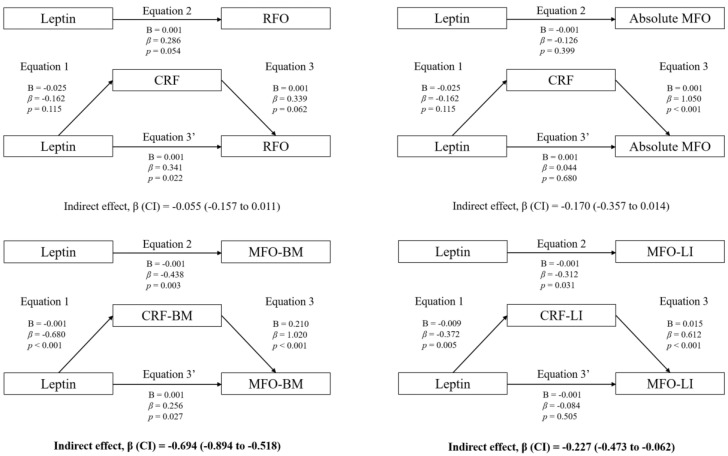
Mediation analysis of CRF in the association between leptin concentration and fat oxidation.

**Figure 3 nutrients-15-02628-f003:**
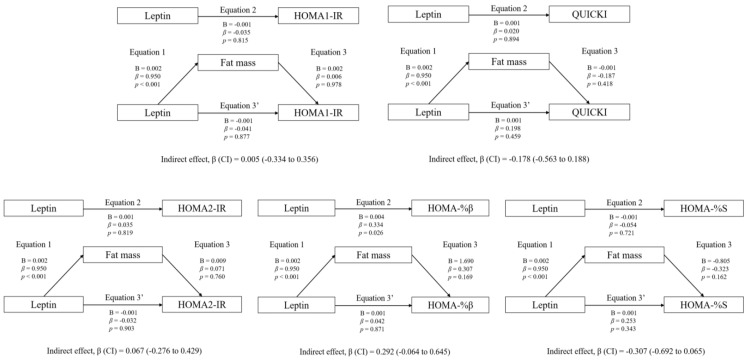
Mediation analysis of fat mass in the association between leptin concentration and insulin resistance/sensitivity.

**Figure 4 nutrients-15-02628-f004:**
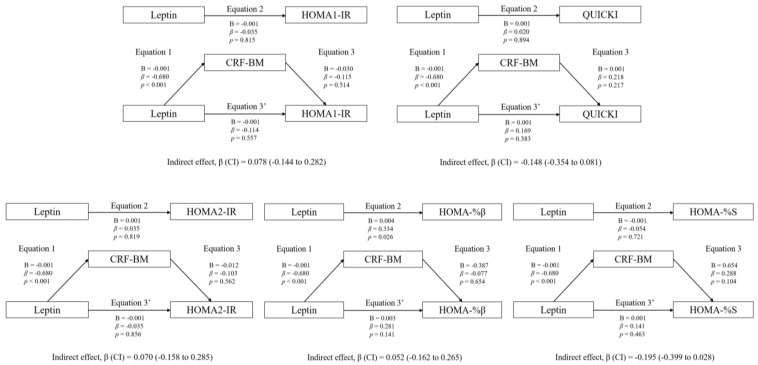
Mediation analysis of CRF in the association between leptin concentration and insulin resistance/sensitivity.

**Table 1 nutrients-15-02628-t001:** General characteristics of total sample and differences between men and women.

	Total (*n* = 65)	Men (*n* = 42)	Women (*n* = 23)	*p*	d
Age (years)	22.55 ± 4.30	22.29 ± 3.79	23.04 ± 5.17	0.502	−0.17
Height (cm)	172.58 ± 8.34	176.61 ± 6.30	165.21 ± 6.38	**<0.001**	**1.80**
Body Mass (kg)	75.28 ± 15.30	78.16 ± 14.91	70.01 ± 14.88	**0.039**	**0.55**
BMI (kg·m^−2^)	25.26 ± 4.74	25.01 ± 4.17	25.70 ± 5.70	0.574	−0.14
Total lean mass (%)	73.35 ± 8.48	76.95 ± 6.58	66.76 ± 7.62	**<0.001**	**1.46**
Body fat (%)	22.17 ± 9.27	18.49 ± 7.40	28.86 ± 8.69	**<0.001**	**−1.32**
VO_2_peak (mL·kg^−1^·min^−1^)	41.79 ± 11.18	45.19 ± 10.56	35.57 ± 9.66	**0.001**	**0.94**
Leptin (pg·mL^−1^)	3881.94 ± 4593.64	1993.93 ± 2957.42	7329.59 ± 5084.73	**<0.001**	**−1.39**
Insulin (pg·mL^−1^)	644.25 ± 382.70	678.51 ± 417.50	581.70 ± 308.04	0.333	0.25
Glucose (mg·dL^−1^)	101.62 ± 9.59	103.45 ± 9.21	98.26 ± 9.56	**0.036**	**0.56**
HOMA1-IR	4.67 ± 2.92	5.01 ± 3.17	4.07 ± 2.35	0.225	0.32
HOMA2-IR	2.40 ± 1.36	2.53 ± 1.48	2.16 ± 1.11	0.301	0.27
HOMA-%β	135.28 ± 56.07	134.86 ± 58.12	136.03 ± 53.37	0.937	−0.02
HOMA-%S	53.47 ± 25.40	51.87 ± 25.82	56.41 ± 24.92	0.495	−0.18
QUICKI	0.313 ± 0.022	0.310 ± 0.023	0.317 ± 0.020	0.263	−0.32
RFO (mg·min^−1^)	98.39 ± 27.29	102.30 ± 29.99	91.23 ± 20.17	0.119	0.41
MFO (mg·min^−1^)	378.35 ± 156.20	403.25 ± 171.04	332.86 ± 114.51	0.053	0.46
MFO-BM (mg·kg^−1^·min^−1^)	5.21 ± 2.30	5.33 ± 2.48	4.98 ± 1.96	0.532	0.15
MFO-LI (mg·(kg·m^−2^)^−1^·min^−1^)	6.99 ± 2.93	6.38 ± 2.68	8.09 ± 3.09	**0.023**	**−0.60**

Sex differences (*p* < 0.05) in the Student *t*-test appear in bold. Values are expressed as mean ± standard deviation. Abbreviations: *p*, *p*-value; d, Cohen’s d; β, Beta; BMI, body mass index; VO_2_peak, peak oxygen uptake; HOMA-IR, homeostasis model assessment of insulin resistance; HOMA-%β, Percentage of steady state beta cell function (insulin secretion); HOMA-%S, percentage of insulin sensitivity; QUICKI, quantitative insulin sensitivity check index; RFO, resting fat oxidation; MFO, maximal fat oxidation during exercise; MFO-BM, MFO relativized to body mass; MFO-LI, MFO relativized to the legs’ lean mass divided by the height squared.

**Table 2 nutrients-15-02628-t002:** Associations of leptin concentration with fatty acid oxidation and insulin resistance/sensitivity in men and women.

	Model 1			Model 2			Model 3			Model 4		
	β	R^2^	*p*	β	R^2^	*p*	β	R^2^	*p*	β	R^2^	*p*
	Men (*n* = 42)											
Fatty acid oxidation												
RFO (mg·min^−1^)	0.122	0.015	0.441	0.140	0.020	0.396	0.144	0.015	0.641	0.171	0.078	0.283
MFO (mg·min^−1^)	−0.128	0.016	0.418	−0.047	0.126	0.763	0.007	0.023	0.981	0.008	0.511	0.946
MFO-BM (mg·kg^−1^·min^−1^)	−0.359	0.129	**0.020**	−0.270	0.258	0.065	−0.239	0.136	0.366	0.202	0.675	0.086
MFO-LI (mg·(kg·m^−2^)^−1^·min^−1^)	−0.234	0.055	0.135	−0.138	0.208	0.353	−0.184	0.057	0.441	−0.125	0.329	0.359
Insulin resistance												
HOMA1-IR	0.143	0.021	0.365	0.156	0.023	0.344	0.328	0.033	0.285	0.149	0.021	0.457
HOMA2-IR	0.199	0.040	0.206	0.216	0.044	0.189	0.352	0.048	0.249	0.225	0.041	0.259
HOMA-%β	0.398	0.158	**0.009**	0.430	0.175	**0.007**	0.414	0.158	0.151	0.469	0.167	**0.015**
Insulin sensitivity												
QUICKI	−0.186	0.035	0.238	−0.188	0.035	0.254	−0.160	0.035	0.601	−0.157	0.036	0.431
HOMA-%S	−0.256	0.066	0.102	−0.254	0.066	0.119	−0.099	0.075	0.740	−0.196	0.072	0.317
	Women (*n* = 23)											
Fatty acid oxidation												
RFO (mg·min^−1^)	0.501	0.251	**0.015**	0.584	0.311	**0.007**	0.807	0.296	**0.023**	0.538	0.271	**0.013**
MFO (mg·min^−1^)	−0.103	0.011	0.639	−0.008	0.088	0.972	0.331	0.102	0.384	0.090	0.566	0.559
MFO-BM (mg·kg^−1^·min^−1^)	−0.436	0.190	**0.037**	−0.309	0.330	0.126	0.039	0.300	0.906	0.239	0.650	0.215
MFO-LI (mg·(kg·m^−2^)^−1^·min^−1^)	−0.313	0.098	0.146	−0.217	0.176	0.323	−0.121	0.125	0.710	0.025	0.443	0.896
Insulin resistance												
HOMA1-IR	−0.289	0.083	0.182	−0.282	0.084	0.227	−0.354	0.085	0.356	−0.515	0.135	0.094
HOMA2-IR	−0.210	0.044	0.335	−0.216	0.045	0.360	−0.363	0.056	0.352	−0.466	0.110	0.133
HOMA-%β	0.164	0.027	0.454	0.122	0.042	0.603	−0.278	0.122	0.457	−0.130	0.114	0.667
Insulin sensitivity												
QUICKI	0.268	0.072	0.216	0.267	0.072	0.254	0.380	0.078	0.324	0.597	0.181	**0.049**
HOMA-%S	0.187	0.035	0.393	0.191	0.035	0.421	0.384	0.054	0.326	0.559	0.175	0.065

Statistically significant results (*p* < 0.05) in the linear regression appear in bold. Models 3 and 4 were relativized in the same way as the dependent variable. Abbreviations: *p*, *p*-value; β, standardized coefficient; R^2^, coefficient of determination; RFO, resting fat oxidation; MFO, maximal fat oxidation during exercise; MFO-BM, MFO relativized to body mass; MFO-LI, MFO relativized to the legs’ lean mass divided by the height squared; HOMA-IR, homeostasis model assessment of insulin resistance; HOMA-%β, percentage of steady state beta cell function (insulin secretion); HOMA-%S, percentage of insulin sensitivity; QUICKI, quantitative insulin sensitivity check index. Model 1: unadjusted; model 2: adjusted for age; model 3: adjusted for body fat mass; model 4: adjusted for cardiorespiratory fitness.

## Data Availability

The data presented in this study are available on reasonable request from the corresponding author.

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
