# Peer review of "Influence of Gender on Plasma Leptin Levels, Fat Oxidation, and Insulin Sensitivity in Young Adults: The Mediating Role of Fitness and Fatness"

_nutrients, 2023, doi:10.3390/nu15112628_

Round 1

Reviewer 1 Report

In the article entitled Influence of gender on plasma leptin levels, fat oxidation, and insulin sensitivity in young adults: The mediating role of fitness and fatness, the authors investigate the relationship between plasma leptin levels and fat oxidation, insulin sensitivity, and cardiorespiratory fitness (CRF) in young adults, considering sex differences.

The topic of the article is very relevant and interesting, but all the abbreviations make it very difficult to read. The images are of poor resolution and therefore unreadable.

Below I have listed a few of my inquiries and additional comments for the article.

Line 23-24: Only data for females are presented, data for males are missing.

Line 81-87: Permit numebr is missing.

Line 89-90: Only data for females are presented, data for males are missing. What was the reason for utilizing a sample with a significant numerical disparity?

Line 109-127: Lack of description of homa indexes.

Line 134-136: Sentance not clear.

Figure 1 and 2: The resolution is low. Images are unreadable.

Author Response

A point-by-point response to the reviewer's comments have been uploaded as a Word file attached.

Thank you

Reviewer 2 Report

Dr. Montes-de-Oca-Garcia and his colleagues conducted a cross-sectional study to explore the relationships between plasma leptin levels and resting fat oxidation (RFO), maximal fat oxidation during exercise (MFO), and insulin sensitivity. They specifically considered the gender differences in these associations and examined the potential mediating effects of fatness and cardiorespiratory fitness (CRF). Their study included 65 young adults, with 23 females and an average age of 22.5±4.3 years and a mean Body Mass Index of 25.2±4.7 kg·m-2. Significant associations between plasma leptin levels and fat oxidation were noted as well as insulin secretion/sensitivity, with varying responses observed within each gender group. Furthermore, they found that the relationship between leptin and MFO was mediated by CRF. While their approach holds promise, there are several issues that currently hinder the manuscript's publication in its present form.

Introduction

 1. Minor- In the introduction (lines 48-54), the authors acknowledged the role of leptin in regulating energy expenditure, carbohydrate metabolism, and weight gain mitigation. They also highlighted the association between obesity (elevated BMI values) and impaired sensitivity of leptin receptors, resulting in increased levels of leptin secretion. Furthermore, they could succinctly mention how leptin promotes satiety and reduces caloric intake by either stimulating Pro-opiomelanocortin (POMC) neurons in the arcuate nucleus (ARC) of the hypothalamus or inhibiting their inhibitors (NPY/AgRP neurons), leading to appetite suppression. They could use as a reference the review of Negrea and his colleagues (2021) or any other title in this regard:

 Negrea, M.O.; Neamtu, B.; Dobrotă, I.; Sofariu, C.R.; Crisan, R.M.; Ciprian, B.I.; Domnariu, C.D.; Teodoru, M. Causative Mechanisms of Childhood and Adolescent Obesity Leading to Adult Cardiometabolic Disease: A Literature Review. Appl. Sci. 202111, 11565. https://doi.org/10.3390/app112311565

2. Minor-Lines(72-73) Knowledge gap should be properly addressed citing more relevant titles, in addition to [ref-14].

Materials and Methods

Major -How did the authors set their inclusion criteria? Are there any references in the literature in this respect? A short paragraph describing the rationale should be added to detail their methodology for selecting the subjects.

Major- Calculating sample size for MFO variable typically require additional information beyond the statistical power, effect size, and significance level. The authors should further explain if they considered the anticipated difference in means or proportions, the variability within groups, and other relevant parameters specific to the study design and research question. These values most probable were be based on their prior research [5,15,17].

Results

Minor-To enhance clarity and achieve higher resolution, it is recommended to split both Figure 1 and Figure 2 into two separate figures. This division will allow each individual figure to have better resolution and visual details.

Discussions

Major- Study limitations should include the acknowledgement of the unbalanced male-to-female ratio, which was roughly 2:1 (1.82 to be more precise). The presence of unbalanced proportions has the potential to impact the generalizability and representativeness of the study findings. Consequently, caution should be exercised when interpreting the proposed linear models, and it is advisable for the authors to explicitly state this limitation.

English – you should properly check the manuscript for typos(e.g unknow…lines 19, 69)

English – you should properly check the manuscript for typos(e.g unknow…lines 19, 69)

Author Response

(The authors gave the same response as above.)

Round 2

Reviewer 2 Report

The authors adequately addressed the raised concerns. Based on the comprehensive improvements made, I highly endorse the publication of this manuscript.